# Are People with Obesity Attracted to Multidisciplinary Telemedicine Approach for Weight Management?

**DOI:** 10.3390/nu14081579

**Published:** 2022-04-11

**Authors:** Luisa Gilardini, Raffaella Cancello, Luca Cavaggioni, Amalia Bruno, Margherita Novelli, Sara P. Mambrini, Gianluca Castelnuovo, Simona Bertoli

**Affiliations:** 1Obesity Unit—Laboratory of Nutrition and Obesity Research, Department of Endocrine and Metabolic Diseases, IRCCS Istituto Auxologico Italiano, Via Ariosto 13, 20145 Milan, Italy; r.cancello@auxologico.it (R.C.); cavaggioni.luca@gmail.com (L.C.); am.bruno@auxologico.it (A.B.); m.novelli@auxologico.it (M.N.); simona.bertoli@unimi.it (S.B.); 2Laboratory of Metabolic Research, S. Giuseppe Hospital, Istituto Auxologico Italiano, IRCCS, 28824 Piancavallo, Italy; s.mambrini@auxologico.it; 3International Center for the Assessment of Nutritional Status (ICANS), Department of Food, Environmental and Nutritional Sciences (DeFENS), University of Milan, 20133 Milan, Italy; 4Psychology Research Laboratory, Istituto Auxologico Italiano IRCCS, 28824 Piancavallo, Italy; gianluca.castelnuovo@auxologico.it; 5Department of Psychology, Catholic University of Milan, 20123 Milan, Italy

**Keywords:** obesity, lifestyle intervention, telemedicine, COVID-19 pandemic

## Abstract

The forced isolation due to the COVID-19 pandemic interrupted the lifestyle intervention programs for people with obesity. This study aimed to assess: (1) the behaviors of subjects with obesity towards medical care during the pandemic and (2) their interest in following a remotely delivered multidisciplinary program for weight loss. An online self-made survey addressed to subjects with obesity was linked to the official website of our institute. Four hundred and six subjects completed the questionnaire (90% females, 50.2 ± 11.6 years). Forty-six percent of the subjects cancelled any scheduled clinical assessments during the pandemic, 53% of whom had chronic disease. Half of the subjects were prone to following a remotely delivered lifestyle intervention, especially with a well-known health professional. About 45% of the respondents were favorable towards participating in remote psychological support and nutritional intervention, while 60% would practice physical activity with online tools. Male subjects and the elderly were more reluctant than those female and younger, especially for online psychological support. Our survey showed an interest on the part of the subjects with obesity to join a multidisciplinary weight loss intervention remotely delivered. Male subjects and the elderly seem less attracted to this intervention, and this result highlights that, even with telemedicine, the approach to weight management should be tailored.

## 1. Introduction

Coronavirus disease 2019 (COVID-19) is a severe acute respiratory syndrome caused by SARS-CoV-2 that was first discovered in Wuhan, China in December 2019 and rapidly spread to the rest of the world [1]. The infection is highly transmissible, and the number of those infected with COVID-19 has now reached more than 364 million patients and over 5,500,000 deaths. 

To contrast and contain the pandemic, at the beginning of March 2020, the Italian Government adopted restriction measurements consisting of a temporary closure of all nonessential activities, strengthening the measures aimed to increase personal hygiene, symptom monitoring, early diagnosis, and patient isolation [2]. The lockdown was repeated in November 2020 and March 2021. All these restrictions required individuals to stay at home, leading to modifications in lifestyles and daily life habits, especially for those in frail categories, such as subjects with obesity. In this context, people are prone in buying large quantities of ultra-processed, unhealthy food to cope with fear, boredom, or anxiety evoked by the worldwide pandemic [3,4]. Moreover, in this difficult situation, individuals with eating disorders may be at a high risk of relapsing or of a worsening of the severity of their disorder [5,6]. Combined with a decrease in the levels of physical activity registered [7], the impaired nutritional habits could lead to weight gain. An Italian study in a small cohort of individuals with obesity showed a significant weight gain 1 month after the beginning of the lockdown period [8]. The increased risk of weight gain during lockdown and the evidence that any degree of obesity has been associated with poor prognosis in patients with SARS-CoV-2 infection [9] have pointed out the importance of ongoing support for obese subjects to manage the disease during the pandemic. Indeed, in Italy, all non-urgent medical visits, including clinical practices planned for obesity management, were deferred to ensure social distancing and reduce the virus spread. Since the resumption of clinical activities after the lockdowns, the fear of contracting the infection in healthcare places, such as hospitals and clinics, has grown in people, and there is still a significant reduction in access to clinical care programs, including metabolic rehabilitation for obesity. The global COVID-19 pandemic has led to a revolution in many fields, promoting alternative strategies with the use of technology. In addition to this, the medical sector has also promoted the usefulness of telehealth and telemedicine, but benefits and barriers in using technology should be considered when dealing with patients [10]. It is worth noticing that telemedicine may represent a novel, effective option in obesity management. In fact, it has previously been demonstrated that video visits with physicians and dieticians can be effective in driving weight loss compared to the standard care [11,12]. IRCCS Istituto Auxologico Italiano (https://www.auxologico.it/, accessed on 10 March 2022) is a specialized national center for obesity care. In Italy, the prevalence of obesity was estimated as 10.9% in 2019, and it is higher in men (11.7%) than in women (10.3%), with increasing prevalence from the north to the south of the country [13]. In our institute, we conduct a 3-month multidisciplinary program aimed at weight loss in obesity-suffering subjects with the involvement of dieticians, physicians, psychologists, and exercise physiologists. The intervention includes individual interviews with the health professionals, nutritional/psychological group sessions, and a one-hour session of moderate intensity physical activity under the supervision of a physical trainer [14]. At the end of the rehabilitation, patients were given an appointment for the regular three months of follow-up visits. During the pandemic, we were forced to stop this rehabilitation program, and we wondered what the barriers were for the patients with obesity to perform the intervention with telemedicine. We conducted a self-reported online survey among the newsletter readers of Istituto Auxologico Italiano from October 2020 to March 2021 to observe the perception of telemedicine of patients with obesity.

The purpose of this survey was to investigate (1) the behaviors of obese subjects towards medical care during the pandemic and (2) their opinion on the possibility of following a remotely delivered program for weight loss, including psychological, nutritional, and physical activity interventions. 

## 2. Materials and Methods

The present study is a cross-sectional design carried out using an online self-made questionnaire (from October 2020 to March 2021) adopting a Google online survey platform (Google LLC, Mountain View, CA, USA). A link to the electronic survey is present on the official website of the IRCCS Istituto Auxologico Italiano (www.auxologico.it, accessed on 10 March 2022) and was shared via the local institute newsletter. Registration to receive the newsletter is open to everyone, but usually, the subscribers are patients who use the services of our institute for healthcare. All participants were requested to provide informed consent through an appropriate checkbox in the survey regarding research purposes. Participants’ answers were anonymous, in accordance with Google’s privacy policy (https://policies.google.com/privacy?hl=it, accessed on 20 February 2022). Each participant was identified by a progressive anonymous number. The self-made survey included a questionnaire composed by 34 questions broken down into three sections: (1) personal anonymous data (age, gender, zip code, education, current work, presence of chronic diseases, availability of an IT tool, weight, and height); (2) disease management during the pandemic (we asked if the patient missed scheduled control visits for any health issue, if they contacted the doctor, and by what means); and (3) obesity management during the pandemic and opinion on the use of telemedicine (whether the patient is prone to following a nutritional, psychological, and physical activity program remotely, what is the best modality, and what are the critical issues). 

The text of the questionnaire is linked as an annex (Appendix A). The inclusion criteria were body mass index values ≥ 30 kg/m^2^ and ages ≥ 18 years. Participants with lower BMI values were excluded by statistical analysis. The participants were asked multiple- or single-choice questions or questions whose answers required the interviewee to enter numeric data. For example: Enter your weight.

The study was conducted in full agreement with the national and international regulations and the Declaration of Helsinki (2000). Participants independently completed an anonymous online questionnaire, explicitly agreeing to participate in the survey. Participants’ personal information were made anonymous to maintain and protect confidentiality. The anonymous nature of the web survey did not allow us to trace in any way sensitive personal data. Therefore, the present web survey study did not require approval by ethics committee. Once completed, each questionnaire was transmitted to the Google platform, and the final database was downloaded as a Microsoft Excel sheet.

### Statistical Analysis 

Continuous variables were expressed as the mean ± standard deviation (SD) and categorical data as frequencies and proportions. Differences between groups were calculated using the Student’s *t*-test for independent samples and analysis of variance for the comparison of multiple groups. Frequencies were compared using a χ2 test. All analyses were performed using SPSS version 26.0 (SPSS Inc., Chicago, IL, USA). A *p*-value < 0.05 was considered statistically significant.

## 3. Results

In total, 465 subjects completed the entire questionnaire. Fifty-nine subjects were only overweight and were excluded. Table 1 showed the characteristics of 406 obese subjects. The subjects were prevalently female, and the age group with the highest frequency (59.4 percent) was 41–60 years old. The mean BMI was 38.0 ± 6.1 kg/m^2^. Men had more severe degrees of obesity compared to women (40.1 ± 5.8 vs. 37.8 ± 5.8 kg/m^2^, *p* < 0.05), had fewer previous diet attempts (more than three diets: 62.5% vs. 83.7%, *p* < 0.001), and tended to have more obesity-related chronic diseases (52.4% vs. 68.3%, *p* = 0.05) than women. There were no differences in age and education levels between the sexes. The percentage of subjects with obesity-related chronic diseases increased with age (34% in subjects aged <40 years (y), 56% in subjects aged 41–60 y, and 71% in subjects aged >60 y, *p* < 0.0001) but not with the degree of obesity. The degree of obesity, gender, and education level were similar across the age groups. Only four subjects (all females) had no electronic tools, three of whom were >65 years. A sedentary behavior was present in 41%. The most declared activity by physically active subjects was “walking”.

### 3.1. Disease Management during Pandemic

The answers given by the participants to the questions about healthcare during pandemic and their opinions on remote visits are summarized in Table 2. Patients who cancelled or postponed a scheduled medical examination (46%) were similar in age, sex, and degree of obesity compared to those who did not but had more frequent chronic diseases (53.2% vs. 37.8%, *p* < 0.005). Of the subjects who cancelled a visit, fifty-three percent contacted the medical doctor in another way, especially by phone, mail, and the WhatsApp (WhatsApp Inc. 2020, Manlo Park, CA, USA). These subjects had more chronic diseases (57.7% vs. 45.7%, *p* < 0.05) than those who did not contact the doctor. Only 24% of subjects believed that the doctor could understand their state of health well through a video consultation, but the percentage rose to 62% if the subject had already met the therapist in a face-to-face visit. 

### 3.2. Obesity Management and Telemedicine

Fifty-five percent of subjects were open to following a remotely delivered lifestyle intervention, 23% only if the health professional was known, 15.4% were undecided, and 6.6% refused. There was no difference in age, sex, degree of obesity, and having a chronic disease in the four groups (Figure 1). Most patients believed that the cost of the online lifestyle intervention should be provided by the national health system. Forty-five percent of the respondents to the survey felt favorably about participating in remote psychological regular support. Men were more reluctant than women (37.5% of males refused this type of therapy vs. 18% of females, *p* < 0.005). Subjects older than 60 years and with chronic diseases tended to be less disposed toward remote psychological therapy than younger subjects and those without chronic diseases (Figure 2). Most people would prefer a video consultation with the psychologist once a week. Another preferred modality is an online interview as needed. The minority would like to have two talks a week. An online nutritional intervention would have been accepted by 46% of the subjects. Males and the elderly were more opposed or indecisive towards this intervention than females and those younger (Figure 3). According to most, nutritional therapy should take place by teleconsulting with a dietician once a week. They also welcomed the sending of video conferencing/written materials concerning diet/nutrition/health. Group activities online once a week was the least welcome option. About 60% of subjects would practice physical activity with online tools supervised by a trainer, while 14% are not interested. The subjects who were more predisposed toward the online program were females and those younger (Figure 4). Concerning physical activity, most of the samples in the study believed that the best way to practice physical activity at a distance was real-time online group lessons, once a week, followed by the reception of training tables and one-on-one meetings via the video platform with an exercise physiologist. The deterring factors of the possibility of doing physical activity online were in order of frequency: laziness, no deterrent, the lack of an appropriate space at home, the difficulty of using electronic tools, and the fear of getting hurt. 

## 4. Discussion

Our study is the first investigation detecting habits and behaviors of obese subjects towards medical care during the pandemic and providing the opinions of subjects with obesity of a remotely delivered program for weight loss, including psychological, nutritional, and physical activity interventions through telemedicine. The COVID-19 pandemic has had not only deleterious effects on infected people but also on the non-COVID-19 patients who have not been able to receive the same level of assistance as before. In Italy, from 15 March 2020, outpatient visits were limited to no deferrable ones, while other appointments were postponed or cancelled to conserve resources and reduce the risk of viral transmission. Our survey highlighted that half of the individuals with obesity responders missed all scheduled medical examinations, especially those with chronic diseases. In according with this finding, a questionnaire coordinated by the Italian National Institute of Health and aimed at people over 65 years revealed that 44% of 1200 subjects interviewed were resigned during the pandemic to missing at least one medical examination (or diagnostic test) that they would need [15]. Bonora et al. showed that the number of visits for diabetic subjects performed during the lockdown period was 47.7% lower than in the same month of the previous 2 years and that the reduction of visits was significantly greater for aged type 2 diabetes patients with heavier complication burdens and complex pharmacotherapies than for younger ones with less complicated diabetes [16]. A study conducted in North Carolina (USA) reported that 53% of outpatient cardiology encounters were cancelled in 2020, and individuals who utilized telehealth tended to be younger, with fewer comorbidities, than those cancelled or referred care [17]. In our survey, patients contacted the doctor, especially via email and WhatsApp, indicating that these media are replacing telephone calls. Although about 60% of the survey participants thought that online consulting might decrease the risk of being infected, only 24% believed that the doctor could perceive their health conditions without an in-person visit. The percentage increased if the subject met the health professional during a previous face-to-face visit. Telemedicine could be a huge opportunity for obesity management during the COVID-19 pandemic, but acceptance by subjects with obesity could be a critical issue. Obesity is a disease in which physician–patient communication is fundamental, and the relationship between the patient and the health professional is important for the success of the intervention [18]. 

For this reason, it was important to investigate how remotely delivered interventions could be perceived in these patients and if there were any phenotypes of patients who were more reluctant toward telemedicine. Our study demonstrated that more than half of the subjects with obesity were willing to participate in an online multidisciplinary lifestyle intervention, but familiarity with the therapist was a conditional factor in the greater acceptance of the therapy. When we analyzed the perceptions of the three interventions separately (psychological, nutritional, and physical activity), we found that the subjects were more reluctant toward online psychological and nutritional therapy than a physical activity program. Men and the elderly tended to be less interested to an online intervention than women and those younger. In particular, about 40% of men with obesity refused online psychological therapy. It is known that men remain less likely than women to access psychotherapy or participate in lifestyle modification programs, including weight loss intervention [19,20]. In our cohort, men had a more severe degree of obesity and associated chronic diseases than women, suggesting that men seek help for their condition (in this case, were attracted by a survey on a weight loss program) later than women and when their health is already compromised with other complications. Thus, appealing and innovative approaches that improve their weight status are needed for men. In a study investigating men’s experiences and perspectives regarding social support after bariatric surgery, male patients reported feeling alone and isolated during the weight loss support groups consisting primarily of women, and they preferred online social support [21]. Other studies confirmed a positive and good response by men to telemedicine [22,23]. Despite this, in our study, we found some reluctance by men to participate in an online weight loss program, especially as regards psychological intervention. Since, as already mentioned above, physician–patient communication is fundamental in obesity management, a face-to-face consultant with the healthcare professional is necessary to explain the modalities and benefits of online lifestyle interventions and to decrease men’s distrust of this type of approach. 

Few studies have examined telemedicine weight loss interventions for older people [24]. Telemedicine may have several advantages, eliminating mobility impediments and the risk of COVID-19 infection. Indeed, older obese subjects have the major risk of severe disease and/or death from COVID-19. In our study, about 20% of subjects were >60 years old, and they had more chronic diseases than those younger. They were less attracted to telemedicine, probably because they felt fragile and needed an in-person consultation with the healthcare professional, including to monitor in their presence their health status. Furthermore, although most of them had an electronic tool and the educational level was no different compared to young subjects, it is possible that they had an inability to manipulate the technology, as well as a cognitive impairment issue. 

It is necessary to promote remotely delivered weight loss interventions tailored for older people that overcome technological and cognitive barriers. The responders chose as the best mode for a nutritional intervention an online visit with a dietician, while real-time group lessons were with an exercise physiologist to practice physical activity at home. The online consultation was probably preferred, because it allows to create interactions between users through facial expressions and voice tone, confirming how important the relationship is with a therapist in the nutritional intervention in obesity management. The online physical activity group lessons have an advantage in that the exercise physiologist can give a visual demonstration of the exercises and that group members can support each other. As a matter of fact, telemedicine may represent a novel, effective option when treating obese patients, but the benefits and barriers in using technology should be carefully considered [10].

Our findings suggest that we could use a “hybrid” model for the management of weight loss during the pandemic. From a practical viewpoint, we could speculate on using a face-to-face consultation for the initial evaluation in order to create a strong patient–physician relationship, and the subsequent visits could be performed remotely, interspersed with traditional visits, especially with frail subjects (i.e., men and elderly). Finally, it is plausible to use telemedicine in providing physical activity group lessons.

The limitations of this survey are also worthy of discussion: (1) It was a self-reported survey, and the study group may not be representative of all the population with obesity, because the survey was addressed and diffused through the newsletter of our institute. In fact, although the newsletter was open to everyone, some subjects may have been unintentionally excluded—in particular, those who did not have an electronic tool or the needed knowledge and skills to carry out the online survey. (2) The gender disproportion confirms the lower interest of men in weight loss issues than women. (3) Information on a previous SARS-CoV-2 infection was not investigated in this survey. This point may have played a role in patients’ decisions to accept or refuse a remote management approach. (4) The questionnaire did not investigate the knowledge of the subjects on how to use the technological aspect of telemedicine. Although almost all the interviewees stated that they have a technological tool (smartphone, computer, or tablet), this does not mean that they were able to use it. This is especially true for the elderly and for those with lower education levels. 

## 5. Conclusions

In conclusion, our survey showed an interest on the part of subjects with obesity to join a multidisciplinary weight loss intervention remotely delivered. Even if with caution, given a possible bias during the recruitment, we noticed that men and the elderly were more reluctant than women and those younger to participate in an online nutritional and psychological intervention. This result highlights once again that, even with telemedicine, the approach to weight management should be tailored. As our survey revealed an interest in telemedicine on the part of people with obesity, we can imagine using this approach also in the post-COVID-19 period. In fact, the implementation of telemedicine in obesity care could minimize patient travel time and missed work, expanding the possibility of treatment to a greater number of subjects with obesity in order to sustain higher adherence to lifestyle changes.

## Figures and Tables

**Figure 1 nutrients-14-01579-f001:**
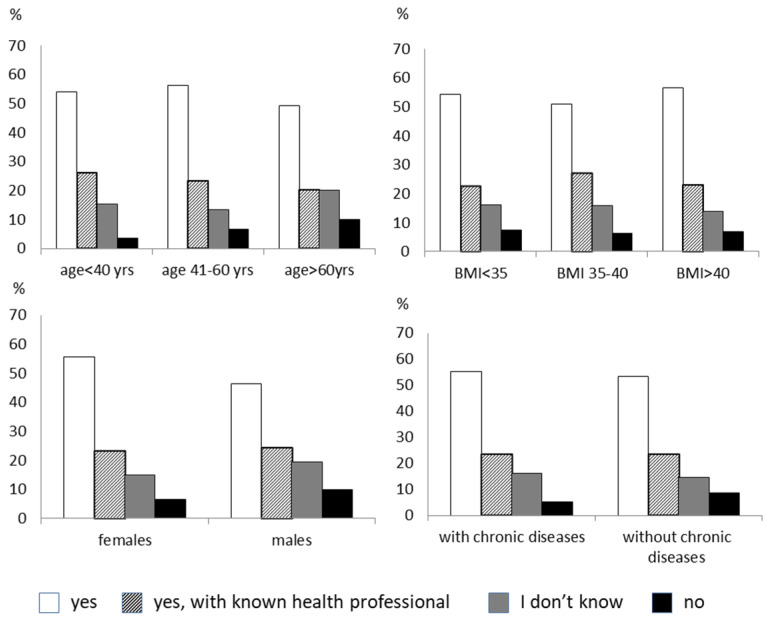
The figure shows the answer to the question: “Would you be like to start an online multidisciplinary intervention for weight management?”. The responders are divided by age, sex, degree of obesity, and the presence of chronic diseases.

**Figure 2 nutrients-14-01579-f002:**
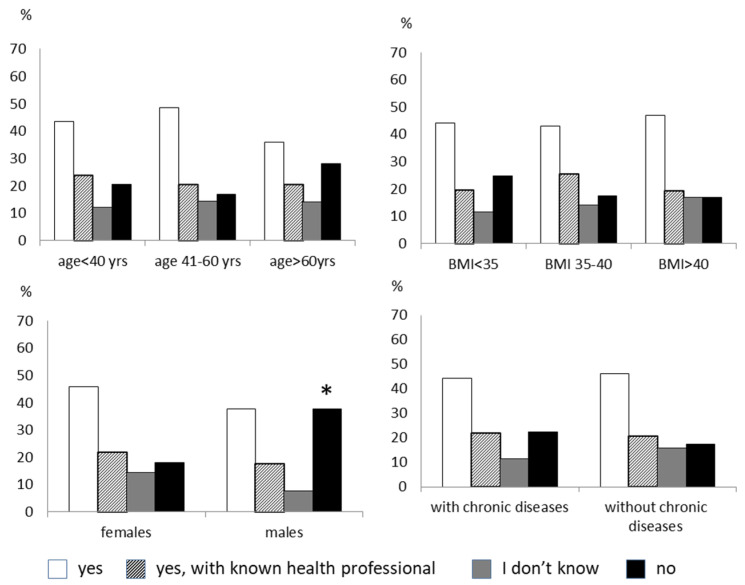
The figure shows the answer to the question: “Would you undergo a remotely delivered psychological intervention?”. The responders are divided by age, sex, degree of obesity, and the presence of chronic diseases. * *p* < 0.005 vs. females.

**Figure 3 nutrients-14-01579-f003:**
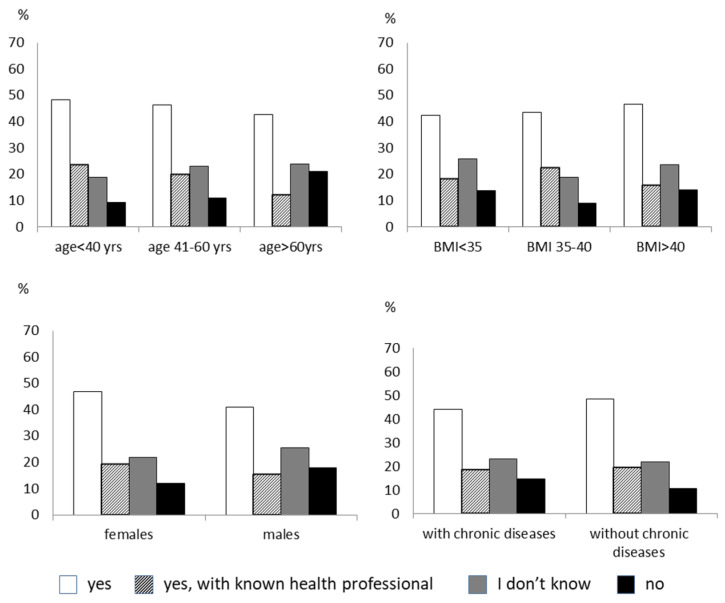
The figure shows the answer to the question: “Would you will join an online nutritional intervention?”. The responders are divided by age, sex, degree of obesity, and the presence of chronic diseases.

**Figure 4 nutrients-14-01579-f004:**
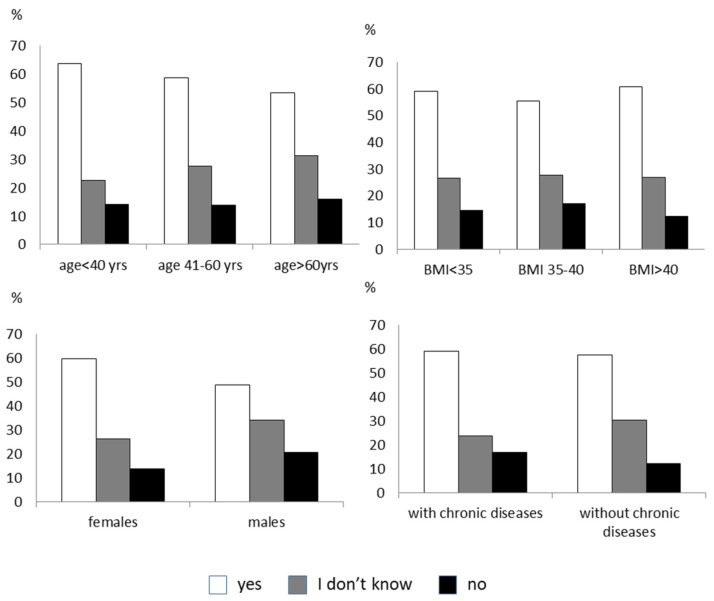
The figure shows the answer to the question: “Would you like to practice physical activity online supervised by an exercise physiologist?”. The responders are divided by age, sex, degree of obesity, and the presence of chronic diseases.

**Table 1 nutrients-14-01579-t001:** Characteristics of 406 subjects with obesity who answered the questionnaire.

*Participant Characteristics*	
Age, years	50.2 ± 11.6
Female, %	90
Educational level	
Primary school, %	5.4
Secondary school, %	57
University, %	37.5
BMI, kg/m^2^	38.0 ± 6.1
Class of BMI	
Class I (BMI 30–34.9)%	36.3
Class II (BMI 35–39.9)%	29.8
Class III (BMI ≥ 40)%	32.3
Subjects with obesity related chronic diseases, %	54.1
Subjects who practise physical activity, %	59
Subjects with at least one diet attempt, %	96
Subjects with at least one electronic tool, %	99

Data are expressed as mean ± SD or percentage (%).

**Table 2 nutrients-14-01579-t002:** Answers given by the participants to questions about medical care during the pandemic with relative percentage (%).

**Have you cancelled or postponed any scheduled clinical assessments during the pandemic?**
No, 53.7%
Yes, 46.3%
**Due to the impossibility of a visit during this period, have you contacted your doctor for the management of your complications in any other way?**
No, never, 47.9%
Yes, by WhatsApp or phone message, 12.7%
Yes, by email, 17.1%
Yes, by telephone, 20.7%
Yes, by video consulting, 1.6%
**Do you think that, during this period, a remote medical video consultation could help you have less health risks?**
No, 12.8%
Yes, 59.0%
I don’t know, 28.1%
**Do you think your doctor can understand your health through a video consultation?**
No, 12.8%
I don’t know, 24.5%
Yes, 24.4%
Yes, but only if he has already met me during a face-to-face visit, 37.7%

## Data Availability

The datasets used during the current study are available from the corresponding author upon reasonable request.

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
