# Peer review of "Are People with Obesity Attracted to Multidisciplinary Telemedicine Approach for Weight Management?"

_nutrients, 2022, doi:10.3390/nu14081579_

Round 1

Reviewer 1 Report

In this manuscript, the authors aimed to investigate the behavior of subjects with obesity   towards medical cares during COVID-19 pandemic as well as their interest to follow a remote program for weight loss. Taken together the results are promising , however the authors are invited consider additional points to support their results:

1) what was the percentage of patients included in the study population with a positive history for COVID-19-Infection. This point is essential and should be included in the statistical analysis since it plays a role in the patient´s decision toward accepting or refusing a remote management approach.

2) According to the results, male subjects and elderly were more reluctant than female and younger, especially for online psychological support. However, the interpretation given by the authors in the discussion part is not convincing and based on results of other studies.
Here the authors are invited to interpret such result basing on the characteristics of Male and older patients in this study. In this context, the authors are invited to demonstrate for example if male and elderly subjects were less obese? or had less educational level ? or less acceptance for technology tools ? which may explain such result. Please add these points in the tables and statistical analysis.

Reviewer 2 Report

There are a couple issues that need to be improved:

  1. the participants/recruitment information is clear. Is it recruited in an institute or anyone on line could participate? It is written that the only inclusion criteria is BMI>30, however, how about age group? I assume you did not intend to include children and adolescents? It is important to provide detailed information on inclusion criteria and exclusion criteria
  2. The results suggest 90% is female, and I am wondering whether there is a bias that most male will not be included, leading to biased results
  3. Given the online filled survey, are there any control or assistant that being provided with the participants on how to answer the questions? The quality on online survey is concern.

Reviewer 3 Report

The article is well written and interesting. Unfortunately, there are some limitations. I have the following suggestions for Authors:

Please describe briefly the epidemiological situation of obesity in Italy in the "Introduction".

The sentences in lines 135-140 are incomprehensible: "Fifty-three per cent of the obese patients contacted the medical doctor during the lockdown. Those who contacted the doctor had more chronic diseases (57.7% vs. 45.7%, p <0.05) than those who did not. Patients who didn’t undergo a scheduled medical examination [...] had more frequent chronic diseases (53.2% vs. 37.8%, p<0.005)."

In my opinion, the main limitation the research methodology is the gender disproportion in the study group with a huge share of women (90%). In the light of this fact, the main conclusion of the authors [I quote: "Male subjects and elderly are more reluctant than female and younger to an online nutritional and psychological intervention and this result highlights that, even with telemedicine, the approach to weight management should be tailored."] does not appear to be objective.

How can your survey results be used to improve care for people with obesity in post-COVID-19 period? Please include your opinion on this in the "Conclusions".

Reviewer 4 Report

This study is very interesting because it deals with the use of telemedicine, an approach that, in view of the saturation of consultations and the limitations of mobility or accessibility to large hospital centers for a large part of the population, could be very useful. However, the reported results present a major limitation: it is necessary to consider the age of the participants, accessibility to the survey, and level of education broadly and include to avoid bias.

For instance,  overweight patients that  can have other comorbidities are often old adults who frecuentely do not have the knowledge and skills to carry out the survey. Accessibility and connectivity also need to be considered.

Round 2

Reviewer 2 Report

I'm wondering the significance and novelty of this study and the impact for future research. What is the major research significance? It seems obvious that telemedicine started to boost during COVID pandemic, including treating noncommunicable diseases such as obesity. What the implications of this study? Are you going to do more tailored intervention among obesity patients via telemedicine or what? It is not clear to me that this study would make a significance contribute to the current knowledge.

Reviewer 3 Report

The authors have appropriately addressed my concerns and suggestions in their revised version

Reviewer 4 Report

From my point of view, I still think that a major limitation of this study is the possible non-inclusion of elderly and digitally illiterate people.  The authors have included this limitation, personally for me it is not enough, but even so I cannot reject the article, because although I do not share their opinion, the authors have justified it.